# Super-Mitobarcoding in Plant Species Identification? It Can Work! The Case of Leafy Liverworts Belonging to the Genus *Calypogeia*

**DOI:** 10.3390/ijms232415570

**Published:** 2022-12-08

**Authors:** Monika Ślipiko, Kamil Myszczyński, Katarzyna Buczkowska, Alina Bączkiewicz, Jakub Sawicki

**Affiliations:** 1Department of Botany and Nature Protection, Faculty of Biology and Biotechnology, University of Warmia and Mazury in Olsztyn, 10-719 Olsztyn, Poland; 2Laboratory of Translational Oncology, Intercollegiate Faculty of Biotechnology, Medicinal University of Gdansk, 80-210 Gdansk, Poland; 3Department of Biology, Institute of Experimental Biology, Adam Mickiewicz University in Poznan, 61-712 Poznan, Poland

**Keywords:** super-barcoding, species identification, liverworts, *Calypogeia*, mitogenome

## Abstract

Molecular identification of species is especially important where traditional taxonomic methods fail. The genus *Calypogeia* belongs to one of the tricky taxons. The simple morphology of these species and a tendency towards environmental plasticity make them complicated in identification. The finding of the universal single-locus DNA barcode in plants seems to be ‘the Holy Grail’; therefore, researchers are increasingly looking for multiloci DNA barcodes or super-barcoding. Since the mitochondrial genome has low sequence variation in plants, species delimitation is usually based on the chloroplast genome. Unexpectedly, our research shows that super-mitobarcoding can also work! However, our outcomes showed that a single method of molecular species delimitation should be avoided. Moreover, it is recommended to interpret the results of molecular species delimitation alongside other types of evidence, such as ecology, population genetics or comparative morphology. Here, we also presented genetic data supporting the view that *C. suecica* is not a homogeneous species.

## 1. Introduction

It is interesting if Paul Hebert and his team, publishing their paper in 2003 [1], thought, at least for a moment, that their research would be groundbreaking and would leave a strong mark on science. The article about the molecular identification of lepidopteran species has reached almost 9000 citations today, and Hebert himself has been called the father of DNA barcoding. The proposed method for species recognition, based on short DNA sequences specific to a species, seemed to be clever and simple. It is indeed so in animals, but in plants, it occurred to be complex, and as some scientists claim, it is like searching for ‘the Holy Grail’ [2]. Whereas in animals the fragment of the ca. 600 bp-long *cox*1 gene delimitates species quite efficiently, in plants, this gene fails because of its low variability [3,4,5,6]. The *cox*1 gene belongs to complex IV of the respiratory chain and encodes subunit 1 of the cytochrome c oxidase. This gene is settled in the mitochondrial genome, which in plants is called peculiar by some scientists [7,8,9]. What amazes them in plant mitogenomes? First of all, its size. In comparison to animals, whose mitogenome is ca. 16 kb in length [7,10], plants have a broad range of mitochondrial genome size, from the smallest (66 kb) in hemiparasitic *Viscum scurruloideum* [11] to colossal, over 11 Mb in *Silene conica* (11.3 Mb) [12] and *Larix sibirica* (11.7 MB) [13]. Quite extreme differentiation in mitogenome size was found also in fungi (1.1–272.2 kb) [14,15]. However, the range of length does not exceed the kb unit. The next striking feature of plant mitogenomes is their variable structure. It is startling that despite the general acceptance of the existence of variable spatial organization of the plant mitogenomes [16,17,18,19,20,21,22,23], still, many publications depict them in ring form without mentioning that it is just a model [9]. Most plant mitogenomes are a combination of linear DNA and some smaller in circular and branched form [10]. This unusual structural plasticity the plant mitogenomes owe to a recombination between numerous repeats scattered throughout the genome [8,24,25,26]. The longer the repeats, the more frequent the recombination. Repeats over 1 kb recombine the most often [11,25], and they are the main cause of structural rearrangements of plant mitogenomes [27], whereas recombination between short repeats may lead to a scramble of the gene order, even among members of the same genus [24,28,29]. The number of genes may also be variable [9,10,30], but it is not the main cause of the large size of plant mitogenomes. The primary reason for this phenomenon is an expansion of their intergenic regions by accumulating sequences from nuclear and chloroplast regions and via horizontal transfer [27,31,32]. By all appearances, plant mitogenomes seem to be very plastic. However, it turns out that their gene-coding sequences evolve slowly, and their synonymous substitution rate is around 100 lower than in animal mitogenomes [28]. Comparing this factor to the nuclear and plastid genome, plant mitochondria are 10 and 3–4 times less variable, respectively [33,34,35]. Plant mitochondria are dissonant: on the one hand, they are very labile in their size, structure and gene content; on the other hand, they are stable in the structure of protein-coding sequences, which is probably the result of the DNA repair mechanism, particularly of homologous recombination [27]. Nevertheless, some exceptions happen, as in Geraniaceae, Plantaginaceae, *Silene* and *Ajuga*, where nucleotide protein-coding sequences evolve faster [36,37,38]. The complete opposite of angiosperm mitogenomes are mitochondrial genomes of the early land plants bryophytes. They are said to be conserved in their form (circular), gene content and order [39,40]. However, it has been shown that the order of genes may be disturbed in some species [41].

Genetic stability of plant mitochondria makes species diagnosis based on these sequences problematic. Therefore, scientists, in the search for DNA barcodes, began to concentrate their attention on the plastid genome, which is more variable [27,33]. The Consortium for the Barcode of Life (CBOL) recommended a two-locus combination of *mat*K+ *rbc*L as a plant barcode [42]. It suggests that species discrimination in plants is more demanding than in animals, and just a single-locus barcode is not enough to accurately identify species. Such simple barcodes are especially prone to incorrect species identification when discriminated species can share haplotypes [43,44]. Taking into account frequent hybridization and introgression processes among plants [43,45,46,47], a lack of a single-locus barcode is not startling. On the other hand, the two-locus barcode (*mat*K+ *rbc*L) recommended by CBOL was successful only in 75% [42]. Therefore, some scientists proposed to use multiple barcodes with application sequences such as *mat*K, *rbc*L, and spacers plastid *trn*H-*psb*A and nuclear ITS [48,49]. There were also suggestions for telling plant species apart by other chloroplast sequences, e.g., *ycf*1 [50,51], *ndh*F [51], *ndh*B, *ndh*H, *trn*T-*trn*L [52,53], *trn*L-*trn*F [54,55] and their combination. Unfortunately, there are cases where even multilocus barcoding does not work. In closely related species, some researchers have proposed the application of an entire chloroplast genome as a ‘super-barcode’ [11], and others have suggested so-called ‘ultra-barcoding’, based on both the whole plastome and nuclear ribosomal DNA [56], because organellar genomes are maternally inherited.

The *Calypogeia* genus, belonging to the family Calypogeiaceae, reaches about 90 species in the world and 10 in Europe, but all of them are tricky in their diagnostics. Morphologically, they are very similar, but the lack of well-developed keys for *Calypogeia* species is problematic. A good diagnostic feature seems to be the presence of oil bodies, but they occur only in fresh plant material. Difficulties also appear when the observed oil body characteristics cannot be matched with a key [57]. Moreover, *Calypogeia* species are characterized by plasticity. Their phenotype is shaped by the environment. It can lead to masking a valid distinction between species [58]. On the other hand, among *Calypogeia* species, some of them are hidden, so-called cryptic species. Although morphologically they look exactly alike, in fact, they form genetically different groups of species. Cryptic species can be found in *C. fissa* [59], *C. sphagnicola* [60,61] and *C. suecica* [52].

In our study, we decided to take up the issue of species identification in the *Calypogeia* genus, one of the most problematic liverwort taxa according to many taxonomists. We undertook this some time ago on the basis of super-barcoding [52]. One issue was unresolved there: the *C. suecica* case. We suggested that it could be a cryptic species and therefore collapsed correct species identification. We decided to study it again in this research.

Having access to the data and already being experienced by our previous research on liverwort mitogenomes [41,62], we also assembled 26 mitochondrial genomes of 11 species of *Calypogeia* to check if there was anything extraordinary there. Surprisingly high variability of this chondriome pushed us to ask ourselves a bold question: is there any possibility to disclose plant species using super-mitobarcoding?

## 2. Results

### 2.1. Analysis of Organelle Genome Variation

The newly assembled organellar genomes of *Calypogeia* are circular and have typical gene order like other liverworts [52,62]. Plastome length ranged from 119,628 bp in *C. arguta* to 120,170 bp in *C. muelleriana*, whereas mitogenomes reached from 159,055 bp in *C. arguta* to 163,324 bp in *C. neesiana*. Genes of the chloroplast genome were located in four regions: a large single copy (LSC), a small single copy (SSC) and two inverted repeat regions (IRs). Although the plastome of *Calypogeia* species is shorter compared to mitogenome, it has more genes: 81 protein coding, 4 rRNAs, 31 tRNAs and 6 *ycf* genes of unknown function (total 122 genes) considering only one copy of IR region. Meanwhile, the mitochondrial genome holds 70 genes: 42 protein-coding genes, 25 tRNAs and 3 tRNAs.

Analysis of two organellar genomes in *Calypogeia* revealed greater variability of the plastome. The nucleotide diversity (π) for this organellar genome (π = 0.04122) was almost twice as high as that of the mitogenome (π = 0.02279). The difference in the variation of organellar genomes is excellently presented in Figure 1, where the blue and red lines indicated the mean π value for the whole mitochondrial and chloroplast genomes, respectively. The most variable 500 bp-long region in the mitogenome of *Calypogeia* lies within the pseudogene *nad*7, and the rest of the five most mutable sliding-window regions are nested within the following spacers: *rrn*5-*rrn*18, *nad*1-*cob*, *nad*2-*rps*12 and *nad*3-*nad*7 pseudogene (Figure 1). On the other hand, considering full-length noncoding regions, the most variable are the following spacers: *rps*8-*rpl*6, *nad*4L-*tatC*, *rps*14-*rps*8, *tatC*-*cox*2 and *atp*9-*rps*2 (π value, Appendix A). Among the coding sequences (CDSs), the most mutable belong to the small ribosomal subunit genes *rps*8 and *rps*1 and to the *rpl*2 gene encoding the large ribosomal subunit, as well as to the ATP synthase gene *atp*4 and the reverse transcriptase gene *rtl* (π value, Appendix A).

In the plastome, the five most changeable 500 bp-long regions also lie within the intergenic areas, in the large single copy (LSC) section. The most variable of them is that within the *psb*A-*ycf*2 spacer. Interestingly, the peaks indicating the most mutable regions within genomes are much sharper in the mitogenome (Figure 1) The highest peak in the mitochondrial genome has a π value of 0.46791 (within the *nad*7 pseudogene), whereas the highest peak π in the plastome is 0.22261 (within the *psb*A-*ycf*2 spacer).

### 2.2. Phylogenetic Analysis

The phylogenetic trees based on both chloroplast and mitochondrial CDSs visibly separated all investigated species of liverworts (Figure 2). *Calypogeia muelleriana* and *C. integristipula* show the highest similarity in both organelle genome sequences among all studied species of the genus *Calypogeia*. As in previous studies [52,63], *Calypogeia arguta* was positioned as the most distant from the other *Calypogeia* species. Plastid-based phylogenetic data show relationships between *Calypogeia* species more precisely, which is supported by posterior probability values reaching one by almost all nodes. Only the relationship between *C. azurea* and *C. azorica* is supported weaker. The same doubtful phylogenetic affinity both between the two mentioned above species and between them and other *Calypogeia* species (*C. sphagnicola* and two groups of *C. suecica*) is indicated by the mitochondrial CDS-based tree. Nevertheless, all *Calypogeia* specimens were also correctly assigned to the species.

### 2.3. Species Delimitation

The results of tree-based analyses were not fully conclusive. The bPTP-ML analysis carried out on the chloroplast ML tree was too sensitive and recognized more species than were taken to the analysis (Table 1).

The GMYC model applied for the mitochondrial coding sequences extended to other liverwort species was more precise. The tree showed only two specimens of *Calypogeia azorica* as two separated species and merged *C. muelleriana* and *C. integristipula* into one species (Figure 3).

Quite similar results were revealed by ASAP (distance-based method) for mitogenome data (Figure 4A). The lowest asap-score at the level of 4.5 had three partitions. One of them showed more species than was expected, because it treated specimens of *C. azorica* as two species. On the other hand, another partition was less sensitive and pointed out two species fusions. The third partition merged only *C. muelleriana* with *C. integristipula* together, but the dendrogram supporting the bar chart indicated that this fusion is dubious. The orange dot on the node between specimens of *C. muelleriana* and *C. integristipula* suggests that such a merger is rather unlikely, and these individuals correspond to two different species. The correctness of the above division is supported by the ASAP analysis based on chloroplast genomes (Figure 4B). The node between *C. muelleriana* and *C. integristipula* is marked with a black dot, which denotes an even lower probability than for mitogenomic data, that these specimens belong to one group. Two partitions with the lowest asap-score values (1.5 and 3.0) generated for chloroplast genomes perfectly distinguished 11 species of *Calypogeia*.

The species delimitation analysis, based on genetic distance conducted in Spider, revealed the existence of a barcoding gap for all 11 tested species of *Calypogeia* (Appendix A). All results were statistically significant. The largest barcoding gaps both for plastome and mitogenome were identified for *C. arguta* and for *C. neesiana*. For these species, the number of molecular diagnostic characters (MDCs) was also the largest: 9720 for *C. arguta* and 894 for *C. neesiana* (Appendix A).

A screening of the mitochondrial genome fragment by fragment with a length of 500 bp revealed some hotspots in genetic variation and indicated nucleotide sequences laying claim to DNA barcodes. It followed that many such places in the mitogenome can be found, which illustrate many peaks on the Spider charts (Figure 5A,E).

## 3. Discussion

### 3.1. Variation of Organelle Genomes

At present, whole-genome sequencing has become one of the fundamental ways of obtaining scientific information. Although more and more mitochondrial genomes are sequenced among plants, still, little is directly written about their genetic variation at the species level [12]. Many studies rather touch on the issue of structural plasticity between closely related species [9,64,65]. Research taking up the subject of intraspecies variation is scarce [12], which raises the problem with an assessment of the degree of variability obtained in one’s own studies. Most papers generally state that a low mutation rate is typical for plant mitogenomes [27], especially for protein-coding regions [65]. It is even more difficult to collect information about the genetic variation of mitogenomes in bryophytes. Most studies report that mitochondrial genomes of bryophytes are invariant, but it refers to their stability at composition and architecture levels [39,40,66]. Among liverworts, the level of genetic variation has been determined only for *Calypogeia* plastomes [52] and for organelle genomes of cryptic species of the *Aneura pinguis* complex [67]. Plastid genomes of bryophytes seem to evolve faster and be slightly more variable than mitogenomes, but they require verification in the future. Currently, it is confirmed only by one study regarding *Aneura pinguis* [67]. However, there are reports stating that the substitution rate of the plastome was three times higher than that of mitogenome in angiosperms and twice as high as the substitution rate of mitogenome in gymnosperms [65]. The greater nucleotide diversity in various regions of plant mitogenome was noticed by van de Paer et al. in their study on the olive family [68]. Our results are in line with the above observations, because the chloroplast genome (π = 0.04122) of the genus *Calypogeia* is twice as variable as the mitogenome (π = 0.02279). Similar disparity values of organelle genome variation (based on π value) were observed in the Oleaceae family. However, it is worth noting that the value of disparateness between mitogenome and plastome in *Calypogeia* is at the species level and in olives at the subspecies (disparity of size 1.5–2 times) and subtribe level (disparity of size 3–5 times) [68]. This may also support the hypothesis that bryophytes evolve more slowly than angiosperms [69], hence a similar level of π-value discrepancy between organelle genomes is shaped at different taxonomic levels as a potential for evolution.

A huge disparity in the nucleotide diversity (π) between organelle genomes was found in the cryptic species complex of *Aneura pinguis* [67]. The nucleotide diversity for the mitochondrial genome in this species was 0.00233 and was almost 22 times lower than in the plastome. In comparison, most π values in the mitochondrial genome for some subgroups of *Oryza sativa* [70] were lower than 0.005, but in the complex of *Olea europaea* (olive) were about 0.031 and about 0.173 in the subtribe Oleinae [68]. So, the π value of 0.02279 for the mitogenome of the genus *Calypogeia* seems to be pretty high. This huge variability of the mitochondrial genome of *Calypogeia* is also noticeable in the number of SNPs (4781) and indels (1068) (Appendix A). The mitogenome of *Aneura pinguis* counted 953 SNPs and 1940 indels [67], whereas in genus *Orthotrichum* s.l., 427 SNPs and 6 indels were found [71]. The species *Scapania ampliata*, belonging like the *Calypogeia* species to leafy liverworts, was detected to have 823 SNPs and 2242 indels in the mitochondrial genome [72]. Meanwhile, species belonging to the complex thalloids showed a much lower level of genetic variation in this organelle genome. There were only 7 SNPs and no indels in *Marchantia polymorpha* subsp. *ruderalis* [73], 12 SNPs and 24 indels in *Dumortiera hirsuta* [74], 18 SNPs and 19 indels in *Riccia fluitans* [75], and 14 SNPs and 7 indels in *Monosolenium tenerum* [40]. Our findings for genus *Calypogeia* and results of other researchers for complex thalloids are in line to some extent with reports that Marchantiopsida (complex thalloid liverworts) show a lower evolution rate than Jungermanniopsida (leafy liverworts) [69,76].

It is not a mystery that noncoding regions in plants evolve rapidly, whereas protein-coding regions evolve leisurely [65,68]. It means that a mutation within the non-coding sequence is more likely. Mitochondrial non-coding sequences of *Calypogeia* have over 6 times more SNPs and almost 50 times more indels than the coding sequences. The average π value for protein-coding regions here was 1.5 times lower than for noncoding regions. In our previous study regarding the variation of the *Calypogeia* plastid genome, the π value for coding and noncoding sequences differed twice with an advantage for the noncoding sequences [52]. The sliding-window analysis indicated exactly the most variable regions in the noncoding sequences (Figure 1). The highest peak of variation in the mitogenome of *Calypogeia* was for *the nad*7 pseudogene. A nonfunctional form of a gene can behave like a noncoding sequence and gather more mutations. The other five most variable mitochondrial regions were located within spacers. The same was true of the plastid genome of *Calypogeia* (Figure 1). The most variable region was the spacer *psb*A-*ycf*2, as we have already reported in our previous study [52].

The general π value obtained in our former research for the *Calypogeia* plastome (π = 0.035076) [52] was slightly lower than in the current study (π = 0.04122), but it was assuredly due to the extension of the number of specimens analyzed in the current study. The variation of the chloroplast genome of the genus *Calypogeia* is outstanding, but the variation of the seemingly stable liverwort mitogenome is striking.

### 3.2. Phylogenetic Relationships

The phylogenetic tree presents *C. arguta* as the most distant from other *Calypogeia* species. This distinctiveness of *C. arguta* was noticed by Bakalin et al., who suggested lumping *C. arguta* and related species into the new genus *Asperifolia* [77]. *Calypogeia muelleriana* and *C. integristipula* show the highest similarity in the mitochondrial genome sequence among all studied species of the genus *Calypogeia* (Figure 2). A close relationship between these species has also been revealed by studies based on the selected cpDNA loci [63] and the whole chloroplast genomes [52]. The latter is also visible in the current research (Figure 2). *Calypogeia muelleriana* and *C. integristipula* are considered distinct species, and they have diagnostic morphological features that allow them to be distinguished [78,79,80]. The species also differ in terms of isozyme and molecular markers [81,82]. Thus, the high similarity in the sequence of mitogenomes revealed in these studies proves the common origin of these two species. These species most likely inherited the organelle genomes from a common ancestor.

In our previous research on species identification based on whole plastomes, we suggested that two specimens of *C. suecica* could belong to two different cryptic species [52]. This study confirmed this hypothesis. We have now divided the species of *C. suecica* into two separate species labeled as Group 1 and 2 and extended the size of each group to three individuals. Both plastid and mitochondrial genomic data indicated these two species groups as genetically distinct (Figure 2). Surprisingly, *C. suecica* Group 1 is more closely related to *C. sphagnicola* than to *C. suecica* Group 2. Our morphological observations point to some differences between the two groups of *C. suecica* [83].

### 3.3. Species Delimitation

The organelle genomes of *Calypogeia* among the rather “evolutionarily lazy” genomes of liverworts were revealed to be unusually variable, which makes them a great foundation for the genetic identification of species. While the DNA barcodes for plants are rather sought among chloroplast sequences [42], the use of mitochondrial sequences for species identification is quite unconventional because of a generally reported lower nucleotide variation of plant mitogenomes [33,34,35]. As determined by CBOL, plastid single-locus DNA barcodes do not cope with plant species delimitation [42]. What is more, even two- or multilocus DNA barcoding fails when plant species are closely related and the sequence variation is insufficient [42,84,85] or taxa are just taxonomically difficult to separate [46,47,86,87]. If in such situations scientists recommend using complete chloroplast genomes as super-barcodes to identify plant species [12], then applying a similar method to much less variable mitogenomes seems reasonable. 

As far as we know, it is the first application of mitogenomes as super-barcodes in plants. We called this method ‘super-mitobarcoding’. Thus far, mitochondrial sequences in plants, in contrast to animals [1], have been useless in identifying species because of their nucleotide stability. The *Calypogeia* case with a high variability level of mitogenomic sequences provides a strong foundation for the idea of super-mitobarcoding. We tested the effectiveness of mitogenomes in *Calypogeia* species using different methods. This approach was recommended by some researchers, especially when closely related species are to be distinguished [88]. Some of the applied methods were not completely successful. The best results were obtained in the Spider analysis based on barcoding gaps. All *Calypogeia* species had a statistically significant barcoding gap and were correctly separated (Appendix A), no matter if it was a mitochondrial or plastid genome. Positive separation results with this method were also obtained in our previous study regarding species delimitation of *Calypogeia* based on classic super-barcoding [52]. The graphs generated in the Spider analysis showed the ‘dynamics’ of the *Calypogeia* mitogenome, where many peaks and dots rise and fall but are densely spaced (Figure 5). For comparison, the plastome was more stable, as our analyses from the year 2020 showed [52]. In the mitochondrial genome can be found nucleotide positions, where the mean K2P distances among species are the greatest (Figure 5A), and the diagnostic nucleotides accumulated in the most number (Figure 5E) or clades are identical (Figure 5D). In the *Calypogeia* plastome, genetic distances between species were at a higher level (most under 0.1) [52] than in mitogenomes (most under 0.02; Figure 5A), but the number of diagnostic nucleotides in various positions of the organelle genome was definitely greater in the mitogenome. On the one hand, this may be due to a more evolutionarily stable mitochondrial genome [65,67]; on the other hand, it may be due to the ‘dynamics’ of the *Calypogeia* mitogenome and high π value of some mitogenomic sites (Figure 1).

It is not surprising that most key positions for species delimitation are within the most variable mitochondrial regions. The greatest genetic distances between species are placed within 500 bp-long fragments of the *nad*7 pseudogene (position ca. 50,000; Figure 5A and Figure 1) and the spacers *nad*1-*cob* (position ca. 90,000–95,000; Figure 5A and Figure 1), *rrn*5-*rrn*18 (position ca. 132,000–134,000; Figure 5A and Figure 1), but also within the most variable protein-coding genes such as *rps*1 (position ca. 65,000; Appendix A) and *rpl*2 (position ca. 55,000; Appendix A). Nevertheless, some regions such as *atp*1-*cox*1 (position ca. 110,000; Figure 5A) or *ccm*B-*ccm*C (position ca. 153,000; Figure 5A), which are not among the most mutable, generate quite a high peak in the genetic distance chart. On the other hand, some genetic distances between species, generated by some of the top variable fragments such as spacers *nad*2-*rps*12 (position ca. 12,000–20,000; Figure 5A) or the *nad*3-*nad*7 pseudogene (position ca. 40,000–48,000; Figure 5A) are lost among others.

Similar observations can be made for the number of diagnostic nucleotides (Figure 5E). Most of them are located within the top variable regions: spacers (*nad*2-*rps*12, *nad*3-*nad*7 pseudogene, *rrn*5-*rrn*18; Figure 1), the pseudogene *nad*7 (Figure 1) and the gene *rpl*2 (Appendix A). On the other hand, quite a lot is located within the spacers *sdh*3-*nad*3 (position ca. 35,000; Figure 5E) and *ccm*FC-*rpl*10 (position ca. 160,000; Figure 5E), which were ranked only 15th and 25th, respectively, among the most variable noncoding sequences (Appendix A). The above observations are congruent with our earlier report showing that sometimes good DNA barcodes are not among the top mutable regions [52]. What is more, a species identification based on the whole chloroplast genome can help with finding in *Calypogeia* single-locus DNA barcodes, which can distinguish species pretty well (e.g., *ndhB*, *ndhH*, *trnT*-*trnL*) [52]. Nevertheless, in the present research, single-locus DNA barcodes were not tested.

The first group of methods for species delimitation we used was based on the application of genetic distances. The methods using a comparison of intra- and interspecific genetic distances are debatable, because they do not appeal to the clear species concept [89]. However, they are widely applied [52,90,91] according to the first suggestion of the barcoding forerunner Hebert [1]. This type of method is also proven to have pretty good results [52,90,92,93]. The assemble species by automatic partitioning (ASAP), based partly on the barcoding gap, gives a possibility of a choice, of which partition is most relevant, prompting the correct answer by the asap-score. Both the Spider analysis (Appendix A) and 1 of 10 ASAP partitions perfectly separated all *Calypogeia* species (Figure 4). Nevertheless, the solution of the ASAP for the plastome was more supported by the lower coefficient asap-score and by the ultrametric clustering tree.

The second group of methods was based on a phylogenetic approach, where all individuals belonging to the same species have a common evolutionary history [94]. Although GMYC and PTP were initially developed for single loci, they are increasingly used for multilocus datasets [95]. GMYC follows a speciation model [96], based on the mitochondrial genome, and almost showed accurate species division of *Calypogeia* (Figure 3). However, the GMYC tree presented both false-negative (*C. muelleriana* and *C. integristipula* as one species) and false-positive (*C. azorica* as two species) results. As discussed above, *C. muelleriana* and *C. integristipula* probably have organelles inherited from a common ancestor. Hence the genetic similarity. However, the oversplit of *C. azorica* into two distinct species is not entirely clear. Because the mitogenome sequences of two representatives of *C. azorica* are very similar to each other (99.9% pairwise identity), which is also visible in the phylogram (Figure 2), it must be a result of GMYC shortcomings. Luo et al. [95] reported that GMYC is vulnerable to the number of loci and individuals, giving false positives. The mitogenomes do not suffer from the number of loci, but in the current research, *C. azorica* was represented only by two specimens (Appendix A). The second deficiency of GMYC is sensitivity to the presence of gene flow [95]. In our dataset, it could have happened between *C. muelleriana* and *C. integristipula*. Hence, the false-negative result for these two species. Considering that some researchers [96,97] reported general overspliting of GMYC, the present results for *Calypogeia* species are quite passable.

The second approach used in the group of phylogenetic-based methods was PTP, but it was applied for plastid genomic data in accordance with a reference [98]. It performed the bPTP-ML version, because it is recommended when interspecies distances are predicted to be small [98], as it is at the low taxonomic level or among closely related species, like in the *Calypogeia* case. bPTP-ML was not very helpful in the present study, because it was too sensitive and indicated too many species (Table 1). Similar to GMYC, the sample size and gene flow can affect the power of the bPTP-ML resolution [95]. Moreover, shallow speciation events may increase the number of false negatives. This could explain the merging of *C. muelleriana* and *C. integristipula* into one species.

The issue of cryptic speciation of *C. suecica*, which we already noted in our previous research from 2020, was resolved. The current analyses, both based on mitogenomic and chloroplast data, definitely divided all specimens of *C. suecica* into two distinct groups: two cryptic species. The present results confirmed the usefulness of every type of DNA barcoding in discovering new species and detecting cryptic speciation [1,49,89]. However, our outcomes supported the scientific suggestion that a single method of molecular species delimitation should be avoided [94], because each method has its own shortcomings, which can be compensated for by the strength of another one. Moreover, it is recommended to interpret the results of molecular species delimitation alongside other types of evidence, such as ecology, population genetics or comparative morphology [99].

## 4. Materials and Methods

### 4.1. Plant Material

*Calypogeia* species used in this study came from the following herbaria: Herbarium of Adam Mickiewicz University (POZW), Herbarium D.A. Callaghan and Herbarium Schäfer-Verwimp (Appendix A). The 26 specimens represent 11 taxa of *Calypogeia*: *C. integristipula, C. fissa, C. sphagnicola, C. paludosa, C. muelleriana, C. azurea, C. arguta, C. azorica* and *C. neesiana. Calypogeia suecica* was divided into two groups marked with numbers 1 and 2 (Group 1, Group 2) as a result of our previous study, where *C. suecica* turned out to be genetically heterogeneous [52]. The total genomic DNA from specimens of each European *Calypogeia* species was extracted using the ZR Plant/Seed DNA MiniPrep™ kit (Zymo Research, Irvine, CA, USA). The species *C. paludosa* and *C. muelleriana* and two groups of *C. suecica* were represented by three specimens, and the rest of the species were represented by two. DNA quantity was estimated using a Qubit fluorometer and the Qubit™ dsDNA BR Assay Kit (Invitrogen, Carsbad, NM, USA).

### 4.2. Organellar Genome Assembly

The genomic library was constructed with the TruSeq Nano DNA kit (Illumina, San Diego, CA, USA) and was sequenced using HiSeqX (Illumina) to generate 150 bp paired-end reads at Macrogen Inc. (Seoul, Korea) with a 350 bp insert size between paired ends. In view of the low amount of available plant material of *C. azorica*, the genomic library had to be constructed using an alternative kit (Qiagen FX library kit), which allows the use of a lower concentration of input DNA. After sequencing, reads were cleaned by removing the adaptor sequences and low-quality reads with Trimmomatic v0.36 [100]. The filtered reads were de novo assembled using the NOVOPlasty 4.3.1 tool [101] and previously published organellar genomes of *Calypogeia* [52,62] as references. Circularized genomes were verified and annotated using Geneious R8 software [102]. Annotated sequences of *Calypogeia* chloroplast and mitochondrial genomes were submitted to GenBank with the accession numbers specified in Appendix A.

### 4.3. Analysis of Organelle Genome Variation

Twenty-six mitochondrial and the same number of plastid genomes of *Calypogeia* species were aligned using the MAFFT plugin [103] within Geneious R8. The second copy of IR was excluded from analyses. The genetic variation and barcoding analyses were conducted using the Spider package [104]. Nucleotide diversity (π) was calculated for each 500 bp-long fragment generated by sliding-window analysis (fragments were generated every 50 bp). The π value, number of SNPs and indels were also calculated for each coding and noncoding region using a custom Python script.

### 4.4. Phylogenetic Analyses

Phylogenetic analyses were performed using mitogenomes and plastomes of 35 species: 26 of the *Calypogeia* genus and 9 other known liverworts. MAFFT software [103] was used to create an alignment of 35 CDS of 42 genes and MrBayes 3.2.6 [105] software was applied for Bayesian analysis. The MCMC algorithm was run for 5,000,000 generations (sampling every 500) with four incrementally heated chains (starting from random trees). Tracer 1.7.1 software [106] was used to determine the number of generations needed to reach stationarity, which occurred at approximately 300,000 generations. Therefore, the first 600 trees were discarded as burn-in, and the remaining trees were used to develop a Bayesian consensus tree. The same methodical approach was applied to plastomes of the 35 above-mentioned liverwort species. In our latest research, we discovered genetic heterogeneity within *C. suecica* [52]. Species delimitation analyses suggested that two specimens of *C. suecica* belonged to two different species. In the current research, *C. suecica* was treated as a species complex and divided into two groups. We added two more specimens to each group in comparison to the previous study. *Marchantia paleacea* was used as an outgroup in both obtained phylogenetic trees.

### 4.5. Species Delimitation

Species delimitation was performed using two different approaches: (1) based on the phylogenetic tree and (2) based on a genetic distance. In the first approach, an input is a tree, and the method assumes that the phylogenetic tree accurately reflects a species differentiation [95]. The generalized mixed Yule-coalescent model (GMYC) [96] was applied to the mitochondrial dataset to delineate species. As an input, the Bayesian tree (obtained earlier in phylogenetic analysis) was used, but previously, the tree was ultraparameterized according to GMYC requirements. Ultrametrization of the tree was performed in the PATHd8 program [107], whereas species identification based on chloroplast genome data was carried out with the use of Poisson tree processes with Bayesian implementation (bPTP) [98]. In comparison to GMYC analysis, bPTP is run on a phylogenetic tree without ultrametrization. In such a case, the length of branches represents the number of substitutions, while an ultrametric tree in the GMYC model presents the stable rate of mutations across all lineages and the vertical branches mirror the time which elapsed between the tree root and the internal nodes. bPTP analysis was carried out on the ML phylogenetic tree (bPTP-ML) obtained using RAxML [108], but in this case, the tree was based on whole plastomes and calculated only for the *Calypogeia* species. bPTP analysis was made with the following settings: unrooted tree; MCMC generations: 500,000; thinning: 100; seed: 123. The bPTP analysis was performed on the server, but GMYC was called directly from the Linux command line (https://github.com/zhangjiajie/pGMYC; accessed on 11 September 2021).

Species delimitation analyses based on genetic distance were carried out in Spider [104], and the assemble species by automatic partitioning (ASAP) [94] was applied. The ASAP generates 10 partitions computing for each partition the asap-score, which is the average of two components: probability of panmixia (*p*-value) and barcoding gap width. The lower the score, the better the partition for the dataset. As input data, alignments of plastomes and mitogenomes of *Calypogeia* species were separately used in each analysis.

## 5. Conclusions

Sometimes, something that looks predictable can surprise. With genomic sequencing data, the stability of the mitochondrial genome of *Calypogeia* was tested. The high nucleotide variability of the mitogenome of the European species of this genus astonished and pushed to check their usefulness as super-barcodes. As it turned out, it might work! The *Calypogeia* species were correctly identified. However, the precision of the species delimitation depended on the applied method and should be interpreted in the light of the general knowledge about the investigated taxon.

The results of the research raise an important issue. Lack of data or scant data on nucleotide variation of genomes among and within bryophyte species may make scientists blind to some lines of research. The science world seems to keep looking at bryophytes as “evolutionary sphinxes of the past” whose organelle genomes evolve slowly. So, can these “sphinxes” surprise with anything else? Fortunately, they can hide many curiosities, but to discover them, you have to think outside the box and not hesitate to check seeming facts.

## Figures and Tables

**Figure 1 ijms-23-15570-f001:**
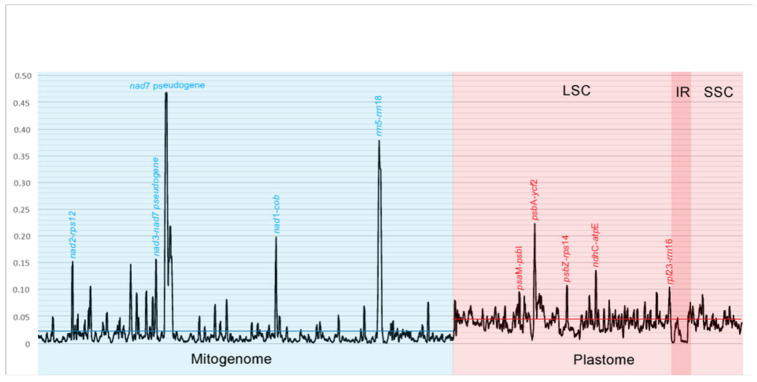
The five most variable regions in plastome and mitogenome of *Calypogeia* obtained by 500 bp-long sliding window. The chart shows the π values.

**Figure 2 ijms-23-15570-f002:**
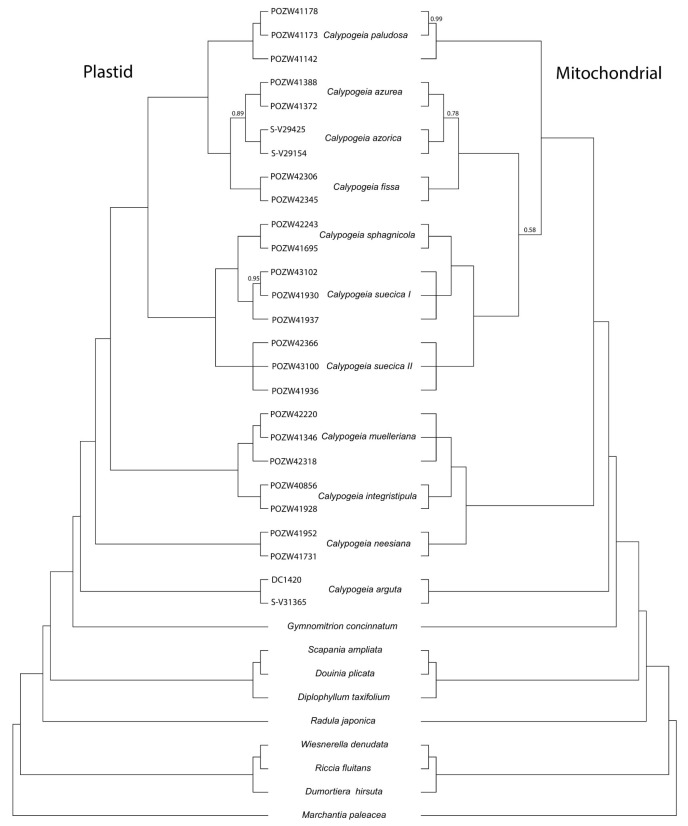
The phylogram based on plastid and mitochondrial CDSs derived from a Bayesian analysis. The posterior probability value lower than 1 is given at the node.

**Figure 3 ijms-23-15570-f003:**
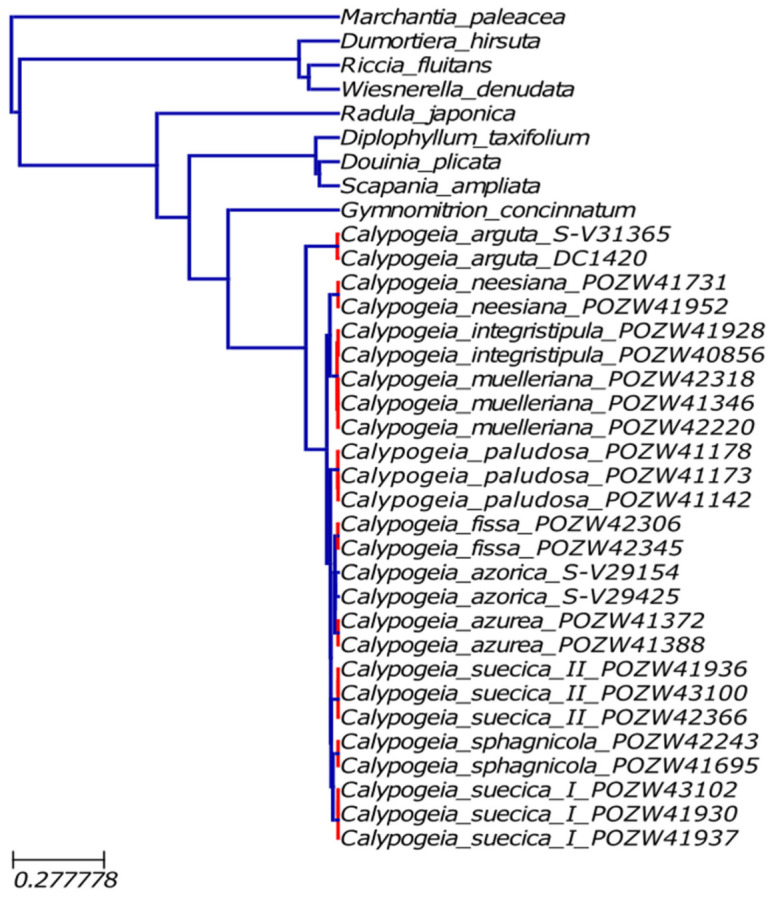
Species delimitation by GMYC analysis. Red lines indicate the same species group.

**Figure 4 ijms-23-15570-f004:**
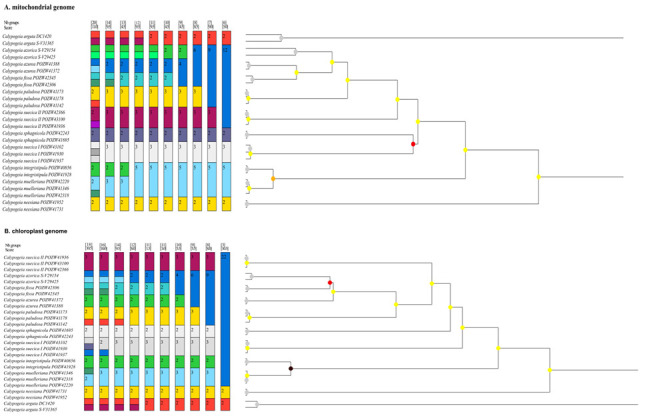
Species delimitation by ASAP analysis for mitochondrial (**A**) and chloroplast (**B**) genomes. The colorful fields indicate the groups of species. Every field contains the number of individuals. Above the colorful bars, the coefficient asap-score (the lower value) and number of species (the upper value) recognized for whole dataset are presented. The cladogram supports the species delimitation presented on the bar charts.

**Figure 5 ijms-23-15570-f005:**
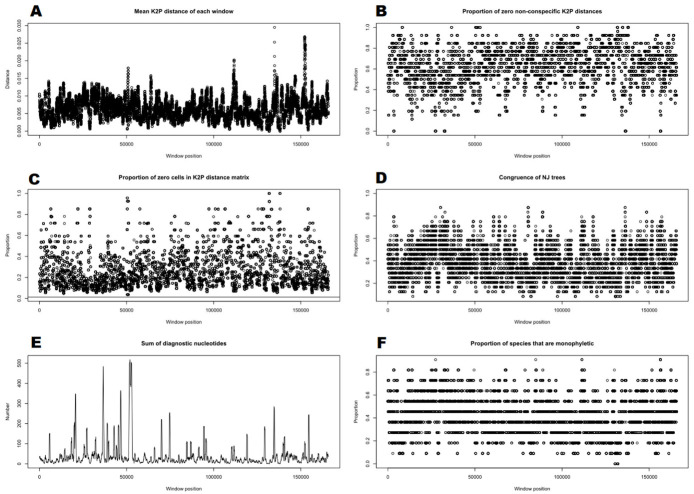
Results of several analyses across the mitochondrial sequences of *Calypogeia* sp. using the sliding-window method. (**A**) The plot of the mean Kimura 2-parameter distance matrix for each 500 bp-long window. (**B**) The proportion of zero non-conspecific distances. (**C**) The proportion of zero cells in the distance matrix. The unbroken horizontal line crossing the *y*-axis at 0 is the proportion of zero cells in the distance matrix created from the full dataset. (**D**) The proportion of clades that are identical between the windows and the full dataset. (**E**) The sum of diagnostic nucleotide positions for all species. (**F**) The proportion of species that are monophyletic.

**Table 1 ijms-23-15570-t001:** Species delimited by bPTP-ML analysis based on chloroplast genomes.

Species Number	Species Name	Bayesian Support (BS)
1	*C. integristipula POZW40856, C. integristipula POZW41928*	0.435
2	*C. fissa POZW42306, C. fissa POZW42345*	0.365
3	*C. paludosa POZW41142*	0.655
4	*C. paludosa POZW41173, C. paludosa POZW41178*	0.330
5	*C. sphagnicola POZW42243, C. sphagnicola POZW41695*	0.417
6	*C. neesiana POZW41952*	0.409
7	*C. neesiana POZW41731*	0.409
8	*C. muelleriana POZW42318*	0.566
9	*C. suecica I POZW41937*	0.543
10	*C. suecica I POZW41930*	0.270
11	*C. suecica I POZW43102*	0.270
12	*C. azorica S-V29425*	0.362
13	*C. azorica S-V29154*	0.362
14	*C. azurea POZW41388*	0.440
15	*C. azurea POZW41372*	0.440
16	*C. muelleriana POZW41346*	0.286
17	*C. muelleriana POZW42220*	0.286
18	*C. suecica II POZW42366*	0.633
19	*C. suecica II POZW43100, C. suecica II POZW41936*	0.321
20	*C. arguta DC1420*	0.402
21	*C. arguta S-V31365*	0.402

## Data Availability

The newly sequenced organellar genomes can be found in GenBank under the accession numbers given in Appendix A. Other supplementary data are also available in the Appendix A.

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
