# Peer review of "Super-Mitobarcoding in Plant Species Identification? It Can Work! The Case of Leafy Liverworts Belonging to the Genus Calypogeia"

_ijms, 2022, doi:10.3390/ijms232415570_

Round 1
Reviewer 1 Report
This is an interesting work that introduces super-metabarcoding as a method toidentify plants using the example of Calypogeia. The research shows that
super-mitobarcoding can also work. The results are well presented. I found some passages in the text that should be improved. These positions are
marked in the attached document. I would like to make the following two comments: 1) Line 23 of the summary says "we also solved the case of Calypogeia suecica
and confirmed it to be a cryptic 23 species." In the taxonomic sense, this
problem cannot be considered to have been solved, since the article does not
redescribe the taxa or describe them in a more detailed morphologically context. In the
added references [55] there are no morphological indications for the
distinction. The attached source [86] represents an unpublished source that
is not freely available. I recommend a more cautious wording here instead of "solved".
2) Interesting is the result in line 171 "Calypogeia arguta was positioned as
the most distant from the other Calypogeia species." Maybe it would be good
to cite here that this species together with some closely related species were
treated in its own new genus by Bakalin et al., Plants 2022, 11(7), 983, as
genus Asperifolia (Warnst.) Troizk., Bakalin, Maltseva.

Author Response
We would like to thank you for doing the review of the manuscript and appreciating the research. Your remarks were valuable and we followed them. We agree with you about cryptic species and that we used too bold word „solve” in the C. suecica context. We have tried to rewrite the indicated sentence in Abstract to give it more cautious tone. We have also referred to the case of C. arguta and its distinctiveness according to your suggestion, but we have done it in Discussion and of course cited the recommended by you research article. We appreciate your effort put into pointing out edidorial mistakes. We corrected all of them together with Table S2 in Supplementary material.
Reviewer 2 Report
The manuscripts are written with good English. However, need to use less of wording we in the sentences. The author need to do minor correction for scientific name format according to ICBN. The conclusion sections need to rewritten and answer the research objective.

Author Response
We would like to thank you for your time to do the review of the manuscript. Your remarks were valuable and we followed them. All species names in the text of the manuscript were written in italics. We have also rewrite Conclusion to make it more relevant to the objective of the research.